# DH-Fusion: Depth-Aware Hybrid Feature Fusion for Multimodal 3D Object Detection

## Abstract

State-of-the-art LiDAR-camera 3D object detectors usually focus on feature fusion. However, they neglect the factor of depth while designing the fusion strategy. In this work, we for the first time point out that different modalities play different roles as depth varies via statistical analysis and visualization. Based on this finding, we propose a Depth-Aware Hybrid Feature Fusion (DH-Fusion) strategy that guides the weights of point cloud and RGB image modalities by introducing depth encoding at both global and local levels. Specifically, the Depth-Aware Global Feature Fusion (DGF) module adaptively adjusts the weights of image Bird's-Eye-View (BEV) features in multi-modal global features via depth encoding. Furthermore, to compensate for the information lost when transferring raw features to the BEV space, we propose a Depth-Aware Local Feature Fusion (DLF) module, which adaptively adjusts the weights of original voxel features and multi-view image features in multi-modal local features via depth encoding. Extensive experiments on the nuScenes dataset demonstrate that our DH-Fusion method surpasses previous state-of-the-art methods w.r.t. NDS. Moreover, our DH-Fusion is more robust to various kinds of corruptions, outperforming previous methods on nuScenes-C w.r.t. both NDS and mAP.

## 1 Introduction

3D object detection has a wide range of applications in the fields of autonomous driving and robotics. A large number of previous works have successfully focused on using a single modality, such as point cloud or images, to design efficient 3D object detectors. However, the performance of these detectors reaches a bottleneck due to the limitations of modality characteristics. For instance, the point cloud modality can only provide rich geometric information while lacks detailed semantic information; the image modality can only provide rich texture information while lacks three-dimensional spatial information. To address the aforementioned issues, we are highly motivated to obtain comprehensive information that represents objects by designing a LiDAR-camera 3D object detector.

In recent years, LiDAR-camera 3D object detection develops rapidly. Some works [1, 4, 28, 33, 67] propose effective methods to integrate information from two modalities at the feature level. However, they all overlook an important factor of depth in their fusion strategies. To understand how point cloud and image information vary with depth, we first conduct statistical and visualization analysis on the nuScenes-mini dataset [3], and find that: (1) The number of points representing objects at near range is relatively large, which allows us to accurately determine the object's location, size, and category, even without the aid of images. As shown in Fig. 1a, there is an average of 163.7 points per object within 0-10 meters, which is a substantial number. We also visualize a car at 6.8 meters in Fig. 1b ① and find it encompasses a considerable number of points, well representing the shape. In contrast, some background noise in the image may interfere with detection (Fig. 1b ②). (2) As the

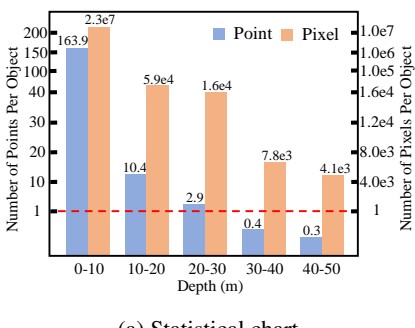
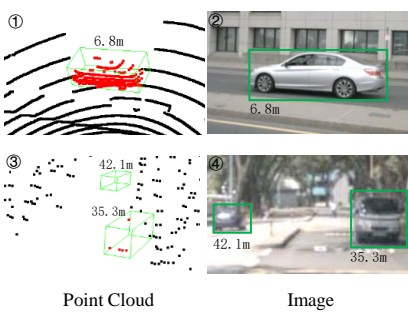

|     |     |
| --- | --- |
| (a) Statistical chart | (b) Visualization |

Figure 1: Statistical and visualization analysis on the nuScenes-mini dataset. (a) The average numbers of points and pixels for each object at different depths. (b) Examples of near-range and long-range objects in images and point cloud. Points within the bounding boxes are colored red for observation.

depth increases, the number of points representing objects decreases rapidly. As shown in Fig. 1a, the number of points within 30-50 meters falls below one per object, meaning that many objects are even not represented by any points, such as the object at 42.1 meters in Fig. 1b ③. In contrast, the complete objects may still be observed on the image, as in Fig. 1b ④, where the image information becomes more important. To address the above problems, we propose a feature fusion strategy that adaptively adjusts the importance of the two modalities based on depth.

Specifically, we propose a novel method for multi-modal 3D object detection, namely Depth-Aware Hybrid Feature Fusion (DH-Fusion). The innovation lies in adaptively adjusting the weights of features by introducing depth encoding to hybrid feature fusion at both global and local levels. The fusion strategy consists of two crucial components: Depth-Aware Global Feature Fusion (DGF) module and Depth-Aware Local Feature Fusion (DLF) module. In DGF, we take point cloud Bird's-Eye-View (BEV) features and image BEV features as inputs, and dynamically adjust the weights of image BEV features based on depth during fusion by utilizing a global-fusion transformer encoder with a depth encoder. To compensate for the information lost when transforming raw features to BEV space, we enhance the fused BEV features at a lower cost by utilizing the original instance features. In DLF, we obtain 3D boxes by utilizing a Region Proposal Network (RPN). Then, the 3D boxes are projected into both LiDAR voxel features and multi-view image features to crop out corresponding local instance features with more detailed information. Afterward, we take these as inputs and dynamically adjust the weights of local multi-view image features and local LiDAR voxel features based on depth through the use of a local-fusion transformer encoder with the depth encoder. In the end, we update local features for each object on the global feature map to enhance the detailed instance information of multi-modal global features for detection.

Our contributions are summarized as follows.

1. We for the first time point out that depth is an important factor to consider while fusing LiDAR point cloud features and RGB image features for 3D object detection. From our statistical and visualization analysis, we can see that image features play different roles as depth varies.

2. We propose a depth-aware hybrid feature fusion strategy that dynamically adjusts the weights of features during feature fusion by introducing depth encoding at both global and local levels. The above strategy can obtain high-quality features for detection, fully leveraging the advantages of different modalities at various depths.

3. Our method is evaluated on the nuScenes [3] dataset and a more challenging nuScenes-C [13] dataset, outperforming previous multi-modal methods and being robust to various kinds of data corruptions.

## 2   Related Work

Since our method is based on conducting 3D object detection using data from multiple modalities, including point cloud and images, we briefly review recent works in the following fields: LiDAR-based 3D object detection, camera-based 3D object detection, and LiDAR-camera 3D object detection.

## 2.1 LiDAR-based 3D Object Detection

LiDAR-based 3D object detectors only take the point cloud as input. Based on their different data representations, they can be divided into point-based [44–46, 64, 65], voxel-based [12, 22, 61, 68, 71], and point-voxel-based [17, 42, 43] methods. The feature extraction networks of point-based methods typically extract features directly from the point cloud through a point-based backbone [40], such as PointRCNN [44]. The voxel-based methods first convert the point cloud into voxels and then extract voxel features through a 3D sparse convolution network [14], such as VoxelNet [71]. Point-voxel-based methods like PV-RCNN [42] combine the above two methods to extract and fuse point and voxel features. The purpose of these approaches is to capture the geometric spatial information of the point cloud. However, point cloud is sparse and incomplete, lacking detailed texture information, which greatly limits the detection performance.

## 2.2 Camera-based 3D Object Detection

Camera-based 3D object detectors only take images as inputs. Depending on the form of inputs, they can be divided into monocular [2, 24, 32, 41, 47, 55], stereo [6, 25, 30, 48, 70], and multi-view [19, 27, 56, 62] 3D object detectors. Early works like FCOS3D [55] input a monocular image and utilize 2D object detectors to directly predict 3D bounding boxes, but these approaches have limited capability in capturing spatial information. Subsequently, stereo and multi-view 3D object detectors are proposed to obtain more precise depth information by constructing spatial relationships among multiple images, such as Stereo RCNN [25] and BEVDet [19]. These methods successfully achieve purely visual 3D object detection, but they do not perform as well as LiDAR-based methods, because the spatial depth information provided by images is not as direct and precise as that provided by point cloud.

## 2.3 LiDAR-Camera 3D Object Detection

LiDAR-camera 3D object detectors take point cloud and images as inputs, and can be classified into early-fusion-based [50, 52, 57, 59, 69], intermediate-fusion-based [1, 4, 28, 33, 67], and late-fusion-based [37, 38] 3D object detectors based on the location of multi-modal information fusion [36].

Early-fusion-based methods perform at the point level, where the typical approach involves enhancing the raw point cloud with semantic information extracted from images. PointPainting [50] and FusionPainting [59] decorate the raw point cloud with semantic scores from 2D semantic segmentation. Similarly, PointAugmenting [52] enhances the raw point cloud using features extracted from a 2D semantic segmentation network. However, early-fusion-based methods are sensitive to alignment errors between the two modalities.

Intermediate-fusion-based methods perform at the feature level. Transfusion [1] first proposes to utilize the transformer for fine-grained fusion from LiDAR BEV features and multi-view image features. FUTR3D [5] encode each modality using deformable attention [73] in its own coordinate and concatenate them for fusion. BEVFusion [28, 33] projects both point cloud and images to BEV space for BEV feature fusion. SparseFusion [58] extracts instance-level features from both two modalities separately, and fuse them to perform detection. Similarly, ObjectFusion [4] utilizes 3D proposals from LiDAR modality to extract instance-level features for fusion. CMT [60] proposes the simultaneous interaction between the object queries and multi-modal features in the transformer encoder and decoder. IS-Fusion [67] proposes feature fusion at both the instance level and scene level. The intermediate-fusion-based methods gradually become a mainstream approach due to the diversity of fusion strategies.

Late-fusion-based methods perform at the bounding box level. Typically, CLOCs [37] obtains 2D and 3D bounding boxes by separately using 2D and 3D object detectors, and then combine them to achieve more accurate 3D bounding boxes. However, the interaction between modalities in late-fusion-based methods is very limited, which constrains model performance.

These multi-modal methods successfully outperform single-modal methods. However, their feature fusion methods do not take depth into account. In contrast, our approach introduces depth information to guide the hybrid feature fusion, boosting the performance of the detector.

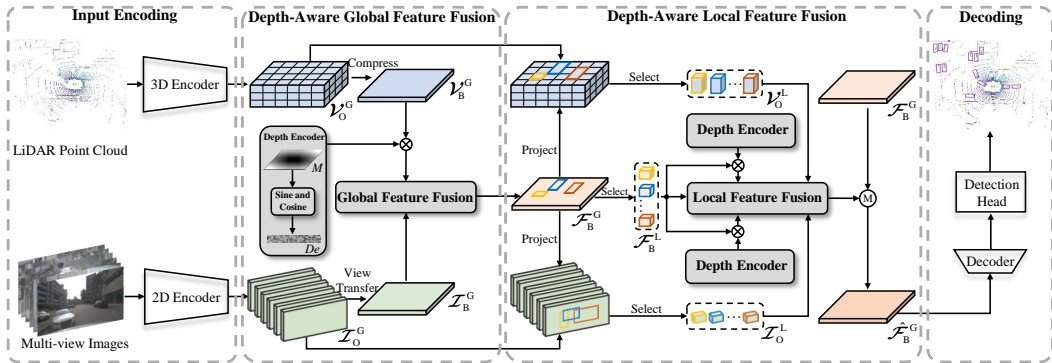

Figure 2: Overview of our method. It introduces depth encoding in both global and local feature fusion to obtain depth-adaptive multi-modal representations for detection. $\otimes$ is the multiplication operation, and $\text{\textcircled{M}}$ is the merge operation.

# 3 Methodology

In this section, we first give an overview of our proposed multi-modal 3D object detector, and then provide a detailed introduction to our proposed feature fusion method.

## 3.1 Overview

We propose a multi-modal 3D object detection method via Depth-Aware Hybrid Feature Fusion (DH-Fusion). As illustrated in Fig. 2, our approach consists of two important feature fusion modules: Depth-Aware Global Feature Fusion (DGF) and Depth-Aware Local Feature Fusion (DLF). In the following, we briefly describe the detection pipeline.

**Inputs.** First, we take the point cloud $P$ and multi-view images $I$ as inputs, where point cloud consists of a set of points: $P = \{P_1, P_2, \cdots, P_{N_l}\}$, and each point has four dimensions: X-axis, Y-axis, Z-axis, and intensity; the multi-view images comprise $N_c$ images: $I = \{I_1, I_2, \cdots, I_{N_c}\}$, each image captured by its corresponding camera.

**Input Encoding.** For the point cloud $P$, we use a 3D encoder to extract raw global voxel features $\mathcal{V}_O^G$; for the multi-view images $I$, we use a 2D encoder to extract image features of all views $\mathcal{I}_O^G$.

**Hybrid Feature Fusion.** Then, for voxel features $\mathcal{V}_O^G$, we compress the height dimension to obtain point cloud BEV features $\mathcal{V}_B^G$; for image features $\mathcal{I}_O^G$, we transform their perspective view to bird's eye view to obtain image BEV features $\mathcal{I}_B^G$. To fully leverage the features from two modalities, we design a DGF module that aims to dynamically adjust the weights of image BEV features based on depth values during feature fusion. Please refer to Sec. 3.2 for more details. To compensate for the information lost when transforming raw features to BEV space, we propose a DLF module that, based on depth, utilizes the raw features to enhance the detailed information of each object instance in global multi-modal features. It consists of three processes: local feature selection, local feature fusion, and merging local features into global features. First, we obtain the local multi-modal BEV features $\mathcal{F}_B^L$, local voxel features $\mathcal{V}_O^L$, and local multi-view image features $\mathcal{I}_O^L$, by cropping the corresponding global features based on the 3D boxes obtained from an RPN; then, it dynamically and individually adjusts the weights of each local feature of $\mathcal{V}_O^L$ and $\mathcal{I}_O^L$ based on depth values during feature fusion; finally, we update local features for each object on the global feature map. Please refer to Sec. 3.3 for more details. In this way, we obtain enhanced multi-modal global features for detection.

**Decoding.** Based on the enhanced multi-modal global features $\hat{\mathcal{F}}_B^G$ that contain rich semantic and spatial information, we utilize a transformer decoder and a detection head to predict the object categories and 3D bounding boxes.

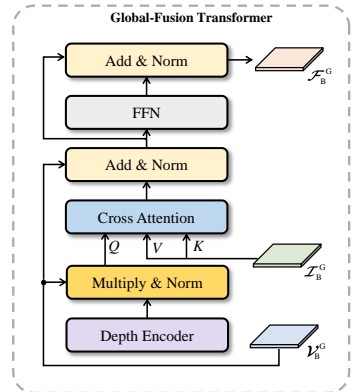

Figure 3: Illustration of the DGF. It consists of a global fusion transformer with the depth encoder.

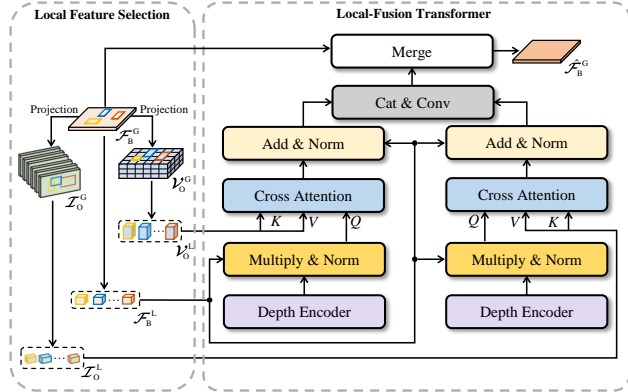

Figure 4: Illustration of the DLF. It consists of a local feature selection module and a local fusion transformer with the depth encoder.

## 3.2 Depth-Aware Global Feature Fusion

As shown in Fig. 3, the DGF module consists of a global-fusion transformer with a depth encoder. In the following, we provide a detailed explanation of each component.

### 3.2.1 Depth Encoder

We introduce depth encoding (DE) in feature fusion to dynamically adjust the weights of image BEV features during fusion. First, we build a depth matrix $M$ to store the depth value of each position element $p_k$ represented as:

$$p_k = \{(x_k, y_k) : d_k\}, k \in [1, n], \tag{1}$$

where $(x_k, y_k)$ are the positional coordinates, $d_k$ is the depth value, and $n$ is the number of elements. Then, we use Euclidean distance to calculate the distance between every element's spatial location $(x_k, y_k)$ and the ego coordinate element's location $(x_{\frac{n}{2}}, y_{\frac{n}{2}})$:

$$d_k = E((x_k, y_k), (x_{\frac{n}{2}}, y_{\frac{n}{2}})), k \in [1, n], \tag{2}$$

where we denote $E(\cdot)$ as the Euclidean distance calculation. The depth matrix $M$ serves as a lookup table to avoid redundant computation of depth values. Since the size of the BEV features is large and the depth distribution is simple, to avoid introducing additional parameters, the depth encoding $De$ is obtained by applying sine and cosine functions [49] to the depth matrix.

### 3.2.2 Global-Fusion Transformer

In the global-fusion transformer, we take the point cloud BEV features $\mathcal{V}_B^G \in \mathbb{R}^{W \times H \times C}$ and image BEV features $\mathcal{I}_B^G \in \mathbb{R}^{W \times H \times C}$ as inputs, and integrate the depth encoding obtained above by multiplying it with the point cloud BEV features, forming the query $Q_{\mathcal{V}}^G = N(\mathcal{V}_B^G \times Conv(De))$, where $Conv(\cdot)$ is a convolution operation to align with the channels of $\mathcal{V}_B^G$, and $N(\cdot)$ is a normalization layer. The image BEV features are queried as the corresponding key $K_{\mathcal{I}}^G$ and value $V_{\mathcal{I}}^G$. We utilize the multi-head cross attention to achieve the interacted feature $\hat{\mathcal{V}}_B^G$ based on depth:

$$\hat{\mathcal{V}}_B^G = CA(Q_{\mathcal{V}}^G, K_{\mathcal{I}}^G, V_{\mathcal{I}}^G), \tag{3}$$

where $CA(\cdot)$ indicates the multi-head cross attention. Afterward, we aggregate the information from both modalities to obtain the fused features $\mathcal{F}_B^G$:

$$\mathcal{F}_B^G = N(FFN(N(\hat{\mathcal{V}}_B^G + \mathcal{V}_B^G)) + N(\hat{\mathcal{V}}_B^G + \mathcal{V}_B^G)), \tag{4}$$

where $N(\cdot)$ is a normalization layer; $FFN(\cdot)$ specifies a feed-forward network containing two convolution operations. In this way, we obtain fused features in which the image features play different roles as the depth varies.

### 3.3 Depth-Aware Local Feature Fusion

As shown in Fig. 4, the DLF module consists of a local feature selection and a local-fusion transformer with the depth encoder. In the following, we provide a detailed explanation of each component.

#### 3.3.1 Local Feature Selection

To compensate for the information lost when transforming point cloud features and image features to BEV space, we enhance the instance details of fused BEV features $\mathcal{F}_B^G$ using instance features from raw voxel features $\mathcal{V}_O^G$ and multi-view image features $\mathcal{I}_O^G$. Specifically, we utilize an RPN to regress $t$ 3D boxes based on the BEV features $\mathcal{F}_B^G$. We directly crop the global fused BEV features $\mathcal{F}_B^G$ based on the regressed 3D boxes to obtain the local fused BEV features $\mathcal{F}_B^L \in \mathbb{R}^{c \times t}$. On the other hand, we project the 3D boxes onto the raw voxel features and multi-view image features to obtain their corresponding local features before global fusion, preserving richer information for each object instance. Specifically, we utilize the voxel pooling operation [12], followed by a 3D convolution operation and a linear layer, to extract local voxel features $\mathcal{V}_O^L \in \mathbb{R}^{c \times t}$; we transform the 3D boxes from bird's eye view to perspective view, and utilize the RoI Align operation [15], followed by a linear layer, to extract instance image features $\mathcal{I}_O^L \in \mathbb{R}^{c \times t}$. By doing this, we obtain the hybrid (before & after global fusion) local features, which will be sent to the subsequent fusion module.

#### 3.3.2 Local-Fusion Transformer

In the local-fusion transformer, the weights of each local raw feature are dynamically adjusted based on depth values during feature fusion, and we update local features for each object on the global feature map. Specifically, we take the local multi-modal BEV features $\mathcal{F}_B^L$, local voxel features $\mathcal{V}_O^L$, and local multi-view image features $\mathcal{I}_O^L$ as inputs, and integrate the depth encoding by multiplying it with the local multi-modal BEV features, forming the query $Q_{\mathcal{F}}^L$. The local multi-view image features and local voxel features are respectively queried as the corresponding key $K_{\mathcal{I}}^L$, $K_{\mathcal{V}}^L$ and value $V_{\mathcal{I}}^L$, $V_{\mathcal{V}}^L$. The two multi-head cross-attention modules are utilized to achieve the interacted features $\hat{Q}_{\mathcal{F}}^L$, $\hat{Q}_{\mathcal{F}}^{L'}$. Note that the computation process of multi-head cross attention is similar to that described in Sec. 3.2.2 and is omitted here. Afterward, we aggregate the above features:

$$\hat{\mathcal{F}}_B^L = Conv(Cat(\hat{Q}_{\mathcal{F}}^L + \mathcal{F}_B^L, \hat{Q}_{\mathcal{F}}^{L'} + \mathcal{F}_B^{L'})), \tag{5}$$

where $Cat(\cdot)$ is the concatenation operation; $Conv(\cdot)$ is used to align with the feature channels of global fused BEV features $\mathcal{F}_B^G$. As a result, we obtain enhanced local features by dynamically calling back rich information in raw modalities at various depths. Afterward, we update the global features $\mathcal{F}_B^G$ by inserting the enhanced local features at corresponding locations.

## 4 Experiments

In this section, we will first introduce the dataset and evaluation metrics, followed by the implementation details. Then, we will compare our method with the state-of-the-art methods on nuScenes and also present results on a more challenging dataset of nuScenes-C with data corruptions. Finally, we will show the ablation studies and qualitative results. More experiments are provided in Appendix A.2.

### 4.1 Experimental Setup

**Datasets and evaluation metrics.** We evaluate our proposed DH-Fusion on the nuScenes benchmark [3] and a more challenging dataset of nuScenes-C [13] with data corruptions. nuScenes dataset provides 700 scene sequences for training, 150 scene sequences for validation, and 150 scene sequences for testing. Each sequence contains 40 frames of 32-beam LiDAR data, and each frame

has six corresponding images covering a 360-degree field of view. It offers calibration matrices that facilitate accurate projection of 3D points onto 2D pixels, and contains 10 object categories that are commonly encountered within autonomous driving. nuScenes-C dataset provides 27 corruptions with 5 severities on the nuScenes validation set, including corruptions at the weather, sensor, motion, object, and alignment level. We use the nuScenes detection scores (NDS) and mean Average Precision (mAP) to evaluate our detection results, where NDS is a comprehensive metric in nuScenes that combines object translation, scale, orientation, velocity, and attribute errors.

**Implementation details.** We implement the proposed DH-Fusion with PyTorch [39] under the open-source framework MMDetection3D [10]. Specifically, for the LiDAR branch, we use VoxelNet [71] with FPN [61] as the 3D encoder. The voxel size is set to [0.075m, 0.075m, 0.1m], and the range of point cloud is [-54m, 54m] along the X-axis, [-54m, 54m] along the Y-axis, and [-3m, 5m] along the Z-axis. For the image branch, we use the ResNet18 [16], ResNet50 [16], and SwinTiny [34] with FPN [29] as the 2D image encoder of DH-Fusion-light, -base, -large, respectively. Correspondingly, the resolution of input images is resized to $256 \times 704$, $320 \times 800$, and $384 \times 1056$. Additionally, we utilize BEVPoolV2 [18] to obtain image BEV features. Following [33], the feature size $W \times H$ is set to $180 \times 180$, the channel $C$ is set to 128, and the channel $c$ is also set to 128. The multi-head cross attention is implemented with 8 heads, and the FFN contains 2 MLP layers with a hidden dimension of 128. Following [58], the number of regressed 3D boxes $t$ is set to 200. More implementation details are provided in Appendix A.1.

## 4.2 Comparison to the State of the Art

Aiming for a fair comparison, we categorize previous methods based on the types of 2D backbones into ResNet50-based, SwinTiny-based, and others, and provide three versions of our proposed method, named DH-Fusion-light, DH-Fusion-base, and DH-Fusion-large. The results are shown in Tab. 1. (1) Compared with the ResNet50-based methods, our DH-Fusion-base outperforms the top method FocalFormer3D [7] by up to 1 pp w.r.t. NDS under the same configuration. Specifically, we reach 74.0% w.r.t. NDS and 71.2% w.r.t. mAP on the validation set, and 74.7% w.r.t. NDS and 71.7% w.r.t. mAP on the test set, while maintaining comparable inference speed of 8.7 FPS on a 3090 GPU. (2) Compared with the SwinTiny-based methods and others, our DH-Fusion-large outperforms the top method IS-Fusion [67] under the same configuration, and runs 2x faster than it. Specifically, we reach 74.4% w.r.t. NDS on the validation set, and 75.4% w.r.t. NDS on the test set, while achieving a faster inference speed of 5.7 FPS on a 3090 GPU, indicating that our proposed method is both more effective and efficient. (3) Furthermore, our DH-Fusion-light surpasses the typical BEVFusion [33] by up to 1 pp w.r.t. all metrics using a lighter 2D backbone, and achieves a real-time inference speed of 13.8 FPS. Overall, our method achieves higher detection accuracy and faster inference speed.

## 4.3 Robustness to Corruptions

We further implement some experiments on the nuScenes-C [13] dataset to evaluate the model's robustness under various corruptions, including changes in weather, data loss or temporal-spatial misalignment in multi-modal inputs, etc. The results for different kinds of corruptions are shown in Tab. 2, and more detailed results for each fine-grained corruption are shown in Appendix A.2.3. We find that our DH-Fusion-light still achieves an average performance of 68.67% w.r.t. NDS and 63.07% w.r.t. mAP under various corruptions, which only decreases by 4.63 pp w.r.t. NDS and 6.68 pp w.r.t. mAP, compared to its performance without corruptions. Performance drop is smaller than that observed with previous methods including BEVFusion [28] across all kinds of corruptions, indicating that our DH-Fusion-light possesses superior robustness. Furthermore, we observe that our DH-Fusion-light is particularly robust against weather and object corruptions, where the performance drop is less than 3pp. The more stable performance indicates that our method is more friendly to practical applications, where data corruption may occur.

## 4.4 Ablation Studies

We conduct ablation studies to first demonstrate the effect of each component of DH-Fusion, then to demonstrate the effect of depth encoding in DGF and DLF, and finally to assess the impact of multiplying depth encoding. All method variants are implemented on the nuScenes validation dataset.

Table 1: Comparisons with the state of the art on the nuScenes `validation` and `test` sets. FPS is measured on a 3090 GPU by default, and * denotes the inference speed on an A100 GPU referred from the original paper. Note that all results are obtained without any model ensemble or test time augmentation.

| Methods | Present at | Image Size - 2D Backbone | FPS | Validation NDS | Validation mAP | Test NDS | Test mAP |
|---|---|---|---|---|---|---|---|
| Image Backbone: ResNet50[16] | | | | | | | |
| Trainsfusion [1] | CVPR'22 | 320 × 800-ResNet50 | 6.5 | 71.3 | 67.5 | 71.7 | 68.9 |
| DeepInteraction [66] | NeurIPS'22 | 448 × 800-ResNet50 | 1.9 | 72.4 | 69.9 | 73.4 | 70.8 |
| MSMDFusion [21] | CVPR'23 | 448 × 800- ResNet50 | 2.1 | 72.1 | 69.7 | 74.0 | 71.5 |
| FocalFormer3D [7] | ICCV'23 | 320 × 800-ResNet50 | 9.2* | 73.1 | 70.1 | 73.9 | 71.6 |
| **DH-Fusion-base (Ours)** | - | 320 × 800-ResNet50 | 8.7 | **74.0** | **71.2** | **74.7** | **71.7** |
| Image Backbone: SwinTiny[31] | | | | | | | |
| BEVFusion [28] | NeurIPS'22 | 448 × 800-SwinTiny | 0.7* | 71.0 | 67.9 | 71.8 | 69.2 |
| BEVFusion [33] | ICRA'23 | 256 × 704- SwinTiny | 9.6 | 71.4 | 68.5 | 72.9 | 70.2 |
| ObjectFusion [4] | ICCV'23 | 256 × 704- SwinTiny | - | 72.3 | 69.8 | 73.3 | 71.0 |
| SparseFusion [58] | ICCV'23 | 256 × 704- SwinTiny | 4.4 | 72.8 | 70.5 | 73.8 | 72.0 |
| IS-Fusion [67] | CVPR'24 | 384 × 1056-SwinTiny | 3.2* | 74.0 | **72.8** | 75.2 | **73.0** |
| Image Backbone: Others | | | | | | | |
| AutoAlignV2 [8] | ECCV'22 | 640 × 1280-CSPNet [51] | 4.8* | 71.2 | 67.1 | 72.4 | 68.4 |
| UVTR [26] | NeurIPS'22 | 640 × 1280-ResNet101 [16] | 1.8 | 70.2 | 65.4 | 71.1 | 67.1 |
| FUTR3D [5] | CVPR'23 | 900 × 1600-VOVNet [23] | 3.3* | 68.0 | 64.2 | 72.1 | 69.4 |
| UniTR [54] | ICCV'23 | 256 × 704-DSVT [53] | 9.3* | 73.3 | 70.5 | 74.5 | 70.9 |
| CMT [60] | ICCV'23 | 640 × 1600-VOVNet | 6.0* | 72.9 | 70.3 | 74.1 | 72.0 |
| UniPAD [63] | CVPR'24 | 900 × 1600-ConvNeXtS [34] | - | 73.2 | 69.9 | 73.9 | 71.0 |
| **DH-Fusion-large (Ours)** | - | 384 × 1056-SwinTiny | 5.7 | **74.4** | 72.3 | **75.4** | 72.8 |
| **DH-Fusion-light (Ours)** | - | 256 × 704-ResNet18 | **13.8** | 73.3 | 69.8 | 74.2 | 70.9 |

Table 2: Robustness experiments on nuScenes-C. Numbers are **NDS / mAP**.

| Methods | Corruption None | Weather | Sensor | Motion | Object | Alignment | Average |
|---|---|---|---|---|---|---|---|
| FUTR3D [5] | 68.05 / 64.17 | 62.75 / 55.51 | 63.66 / 56.83 | 53.16 / 44.43 | 65.45 / 61.04 | 62.83 / 57.60 | 62.82[↓5.23] / 56.99[↓7.18] |
| TransFusion [1] | 69.82 / 66.38 | 65.42 / 59.37 | 66.17 / 59.82 | 51.52 / 41.47 | 68.28 / 64.38 | 61.98 / 54.94 | 63.74[↓6.08] / 58.73[↓7.65] |
| BEVFusion [33] | 71.40 / 68.45 | 67.54 / 61.87 | 67.59 / 61.80 | 55.19 / 47.30 | 68.01 / 65.14 | 63.94 / 58.71 | 66.06[↓5.34] / 61.03[↓7.42] |
| **DH-Fusion-light (Ours)** | **73.30 / 69.75** | **72.19 / 67.48** | **69.16 / 62.87** | **57.07 / 47.52** | **71.01 / 67.11** | **67.24 / 62.38** | **68.67**[↓4.63] / **63.07**[↓6.68] |

**Effect of DGF and DLF.** To demonstrate the effect of DGF and DLF, we conduct experiments by integrating the components one by one into the baseline, BEVFusion [33]. The results are shown in Tab. 3. We find that our DGF improves the baseline performance by 1.0 pp w.r.t. NDS and 0.9 pp w.r.t. mAP. This demonstrates that dynamically adjusting the weights of the image BEV features during fusion is effective for 3D object detection. Additionally, our DLF improves the baseline performance by 1.3 pp w.r.t. NDS and 0.8 pp w.r.t. mAP, which indicates that dynamically adjusting the weights of the local raw instance features based on depth during fusion effectively compensates for the information loss caused by the transformation of global features into the BEV feature space. The results of integrating both components show an improvement of 1.9 pp w.r.t. NDS and 1.3 pp w.r.t. mAP, well verifying the benefits of dynamically fusing global and local hybrid features based on depth.

**Effect of depth encoding in DGF and DLF.** To evaluate the effectiveness of our depth encoding, we conduct experiments where the depth encoding is removed from the DGF and DLF modules, respectively. The results are shown in Tab. 4. When removing the depth encoding from Baseline+DGF, the performance drops by 0.6 pp w.r.t. NDS and 0.4 pp w.r.t. mAP. Similarly, when removing depth encoding from Baseline+DLF, the performance also decreases by 1.1 pp w.r.t. NDS and 0.9 pp w.r.t. mAP. These results indicate that our depth encoding is effective. Furthermore, we observe that removing the depth encoding from the DLF module results in a larger performance drop, suggesting that depth encoding plays a more crucial role in local feature fusion.

**Impact of different operations for depth encoding.** We conduct experiments with different operations of depth encoding, including concatenation, summation, and multiplication. The results in Tab. 5, show that the multiplication operation consistently outperforms the summation and concatenation operations w.r.t. both metrics. The superior performance of multiplication can be attributed to its ability to more effectively modulate the feature maps based on depth information. Unlike summation, which simply shifts the feature values, or concatenation, which increases the dimensionality without direct interaction, multiplication allows for more interaction between the

Table 3: Ablation studies of each proposed module.

| Baseline | DGF | DLF | NDS | mAP |
|:---:|:---:|:---:|:---:|:---:|
| ✓ | | | 71.4 | 68.5 |
| ✓ | ✓ | | $72.4^{\uparrow 1.0}$ | $69.4^{\uparrow 0.9}$ |
| ✓ | | ✓ | $72.7^{\uparrow 1.3}$ | $69.3^{\uparrow 0.8}$ |
| ✓ | ✓ | ✓ | $\mathbf{73.3}^{\uparrow 1.9}$ | $\mathbf{69.8}^{\uparrow 1.3}$ |

Table 4: Ablation studies of depth encoding (DE) in DGF and DLF.

| Methods | NDS | mAP |
|:---:|:---:|:---:|
| Baseline + DGF | 72.4 | 69.4 |
| w/o DE | $71.8^{\downarrow 0.6}$ | $69.0^{\downarrow 0.4}$ |
| Baseline + DLF | 72.7 | 69.3 |
| w/o DE | $71.6^{\downarrow 1.1}$ | $68.4^{\downarrow 0.9}$ |

Table 5: Ablation studies of different operations for depth encoding.

| Methods | NDS | mAP |
|:---:|:---:|:---:|
| Summation | 72.8 | 69.2 |
| Concatenation | 72.5 | 68.7 |
| Multiplication | **73.3** | **69.8** |

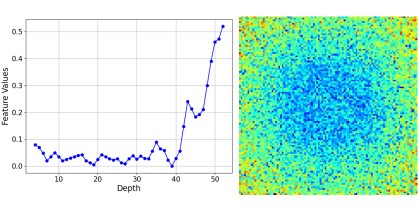

(a) Attention weights  (b) Average map

Figure 5: Attention weights applied on BEV image features in DGF vary with depth.

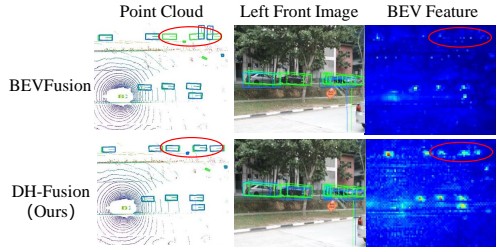

Figure 6: Qualitative detection results and BEV features of BEVFusion and ours. We show the ground truth boxes in green, and the prediction boxes in blue.

depth encoding and features, leading to better feature representation and ultimately improving the detection performance.

### 4.5 Qualitative Results

To better understand how depth encoding affects the feature fusion, in Fig. 5, we plot a curve to observe how the attention weights applied on the image BEV features in our DGF module vary with depth, and visualize the average attention map. It is evident that the weights of the image BEV features stay low in near range, but go up significantly as depth increases when the depth is larger than 40 meters. This trend supports our hypothesis that the image modality would become more important as depth increases. In this way, our depth encoding allows the model to dynamically adjust the weights of image BEV features based on depth.

We also compare the detection results of our DH-Fusion method with the baseline BEVFusion [33] in Fig. 6, where we clearly find that our method better localizes those distant objects compared to BEVFusion. These results demonstrate that our proposed multi-modal fusion strategy based on depth is more effective for detection. Besides, we exhibit the corresponding BEV feature maps, where our method shows a stronger feature response for the foreground objects, especially for distant ones. That is why our feature fusion strategy can provide higher-quality detection results. More qualitative results can be found in Appendix A.3.

## 5 Conclusion

In this paper, we for the first time point out that different modalities play different roles as depth varies via statistical analysis and visualization. Based on this finding, we propose a feature fusion strategy for multi-modal 3D object detection, namely Depth-Aware Hybrid Feature Fusion (DH-Fusion), that dynamically adjusts the weights of features during feature fusion by introducing depth encoding at both global and local levels. Extensive experiments on the nuScenes dataset demonstrate that our DH-Fusion method surpasses previous state-of-the-art methods w.r.t. NDS. Moreover, our DH-Fusion is more robust to various kinds of corruptions, outperforming previous methods on the nuScenes-C dataset w.r.t. both NDS and mAP. Our method uses an attention-based approach to interact with the two modalities, making the detection results sensitive to modality loss. We plan to further explore feature fusion methods that are robust to modality loss. Although our method improves detection performance, emergency plans still need to be implemented in practical applications to ensure personnel safety.

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

# A Appendix

## A.1 Additional Implementation Details

During training, we adopt a one-stage strategy like DAL [20]. The whole pipeline is trained for a total of 20 epochs with the AdamW optimizer [35] loading from the pre-trained weights from the ImageNet [11] classification task only. Meanwhile, we use CBGS [72] to resample the training data, and the one-cycle learning policy with a maximum learning rate of $2.0 \times 10^{-4}$. The batch size is set to 8 on 4 3090 RTX GPUs. We adopt random flipping along both X and Y-axis, the random scaling in [0.95, 1.05], and random rotation in [-$\pi$/8, $\pi$/8] to augment the LiDAR data, and the random rotation in [-5.4°, 5.4°] and random resizing in [-0.06, 0.44] to augment the images. During evaluation, we test a single model without any data augmentation on a single 3090 RTX GPU.

## A.2 Additional Experiments

### A.2.1 3D Multi-Object Tracking Experiments

We evaluate our DH-Fusion on the nuScenes tracking benchmark for 3D multi-object tracking (MOT) task. Following ObjectFusion [4], we adopt the same tracking-by-detection algorithm that uses velocity-based closest point distance matching, which is more effective than 3D Kalman filter [9]. For fair comparisons, we report the results of our DH-Fusion-light capable of real-time detection on the nuScenes validation set, as shown in Tab. 6. We find that our DH-Fusion-light outperforms BEVFusion [33] and ObjectFusion [4] by 2.0 pp and 0.6 pp w.r.t. AMOTA. These results demonstrate that our DH-Fusion provides 3D detection boxes of higher quality, benefiting the downstream task of 3D MOT.

Table 6: Comparisons on nuScenes validation set for 3D multi-object tracking.

| Methods | AMOTA ↑ | AMOTP ↓ | IDS ↓ |
|---|---|---|---|
| TransFusion [1] | 71.8 | 60.3 | 694 |
| BEVFusion [33] | 72.8 | 59.4 | 764 |
| ObjectFusion [4] | 74.2 | 54.3 | 611 |
| **DH-Fusion-light (Ours)** | **74.8** | **50.3** | **539** |

### A.2.2 Evaluation at Different Depths

Since our fusion strategy is depth-aware, it is necessary to validate our method at different depths. Following [4], we categorize annotation and prediction ego distances into three groups: Near (0-20m), Middle (20-30m), and Far (>30m). As shown in Tab. 7, compared to ObjectFusion [4], our DH-Fusion-light consistently improves performance across all depth ranges. Specifically, our method achieves a 47.1 mAP in the long range (>30m), surpassing ObjectFusion by 5.5 pp w.r.t. mAP. These results indicate that our method is more effective across different depths, especially in detecting distant objects.

Table 7: Comparisons on nuScenes validation set at different depths. The numbers are **mAP**.

| Methods | Near | Middle | Far |
|---|---|---|---|
| TransFusion-L [1] | 77.5 | 60.9 | 34.8 |
| BEVFusion [33] | 79.4 | 64.9 | 40.0 |
| ObjectFusion [4] | 79.7 | 65.4 | 41.6 |
| **DH-Fusion-light (Ours)** | **80.3** | **66.5** | **47.1** |

### A.2.3 Detailed Results on the nuScenes-C

We further provide the detailed results of each fine-grained corruption on nuScenes-C in Tab. 8. The results are highly consistent with the average values of each kind of data corruption.

## A.3 More Visualization

As an extension of Fig. 6 in the manuscript, we provide additional examples of 3D object detection results and BEV features from our baseline, BEVFusion [33], and our DH-Fusion. In various samples, our method consistently achieves higher accuracy and recall in 3D detection results, with

stronger feature responses for distant objects compared to BEVFusion. These results demonstrate the effectiveness of the proposed method in dynamically adjusting the weights of features based on depth during fusion at both global and local levels.

Table 8: Comparisons for each corruption level on the nuScenes-C. Corruptions exist in both modalities by default. (L) means that only the point cloud modality has corruptions, and (C) means that only the image modality has corruptions. Numbers are **NDS / mAP**.

| Corruption | | FUTR3D | TransFusion | BEVFusion | **DH-Fusion** |
|---|---|---|---|---|---|
| None | | 68.5 / 64.17 | 69.82 / 66.38 | 71.40 / 68.45 | **73.30 / 69.75** |
| Weather | Snow | 61.52 / 52.73 | 68.29 / 63.30 | 68.33 / 62.84 | **71.47 / 65.98** |
| | Rain | 64.47 / 58.40 | 69.40 / 65.35 | 70.14 / 66.13 | **72.05 / 67.32** |
| | Fog | 61.20 / 53.19 | 62.62 / 53.67 | 62.73 / 54.10 | **72.13 / 67.24** |
| | Sunlight | 63.61 / 57.70 | 61.36 / 55.14 | 68.95 / 64.42 | **73.18 / 69.44** |
| Sensor | Density | 67.58 / 63.72 | 69.42 / 65.77 | 71.01 / 67.79 | **72.94 / 69.15** |
| | Cutout | 66.91 / 62.25 | 68.30 / 63.66 | 70.09 / 66.18 | **71.99 / 67.45** |
| | Crosstalk | 67.17 / 62.66 | 68.83 / 64.67 | 70.72 / 67.32 | **73.23 / 69.55** |
| | FOV Lost | 45.66 / 26.32 | 47.89 / 24.63 | **48.65 / 27.17** | 43.41 / 20.78 |
| | Gaussian (L) | 64.10 / 58.94 | 62.32 / 55.10 | 65.99 / 60.64 | **69.04 / 63.51** |
| | Uniform (L) | 67.28 / 63.21 | 68.68 / 64.72 | 70.18 / 66.81 | **72.54 / 68.79** |
| | Impulse (L) | 67.47 / 63.42 | 69.06 / 65.51 | 70.63 / 67.54 | **72.75 / 68.91** |
| | Gussian (C) | 62.92 / 54.96 | 68.94 / 64.52 | 69.35 / 64.44 | **71.55 / 66.16** |
| | Uniform (C) | 64.43 / 57.61 | 69.33 / 65.26 | 70.06 / 65.81 | **72.46 / 67.99** |
| | Impulse (C) | 63.07 / 55.16 | 68.89 / 64.37 | 69.25 / 64.30 | **71.66 / 66.41** |
| Motion | Compensation | **39.62 / 31.87** | 25.69 / 9.01 | 36.76 / 27.57 | 32.51 / 15.99 |
| | Moving Obj. | 56.41 / 45.43 | 60.03 / 51.01 | 59.42 / 51.63 | **68.12 / 60.62** |
| | Motion Blur | 63.44 / 55.99 | 68.85 / 64.39 | 69.38 / 64.74 | **70.58 / 65.95** |
| Object | Local Density | 67.62 / 63.60 | 69.34 / 65.65 | 70.77 / 67.42 | **72.48 / 68.87** |
| | Local Cutout | 66.45 / 61.85 | 67.97 / 63.33 | 68.11 / 63.41 | **69.62 / 64.17** |
| | Local Gaussian | 66.85 / 62.94 | 67.96 / 63.76 | 68.32 / 64.34 | **71.32 / 67.14** |
| | Local Uniform | 67.92 / 64.09 | 69.67 / 66.20 | 70.68 / **67.58** | **71.34** / 66.03 |
| | Local Impulse | 67.89 / 64.02 | 69.64 / 66.29 | 70.93 / 67.91 | **71.83 / 68.15** |
| | Shear | 61.15 / 55.42 | 66.43 / 62.32 | 62.95 / 60.72 | **68.41 / 65.23** |
| | Scale | 62.00 / 56.79 | 67.81 / 64.13 | 66.00 / 64.57 | **71.40 / 68.90** |
| | Rotation | 63.67 / 59.64 | 67.42 / 63.36 | 66.31 / 65.13 | **71.62 / 68.35** |
| Alignment | Spatial | 67.75 / 63.77 | 69.72 / 66.22 | 71.35 / 68.39 | **71.95 / 69.52** |
| | Temporal | 57.91 / 51.43 | 54.23 / 43.65 | 56.62 / 49.02 | **62.53 / 55.24** |
| Average | | 62.82 / 56.99 | 64.71 / 58.73 | 66.06 / 61.03 | **68.67 / 63.07** |

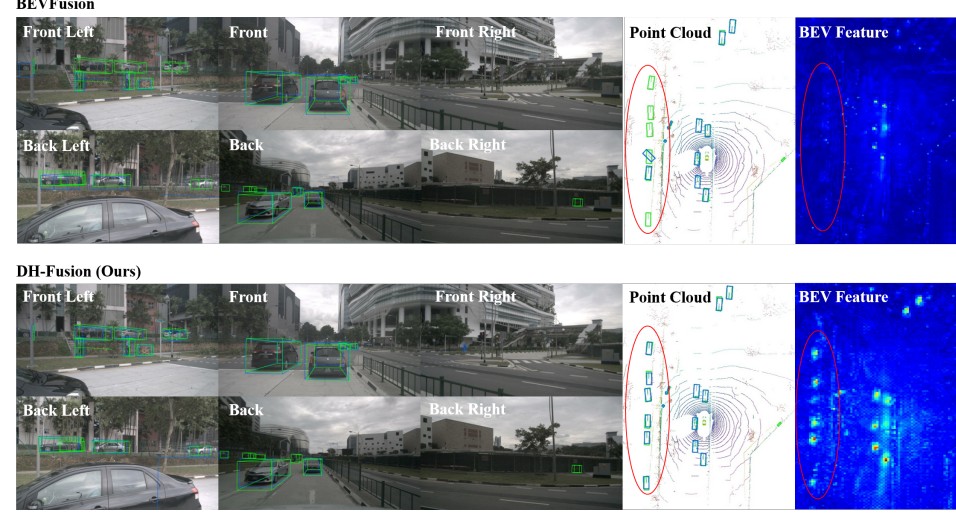

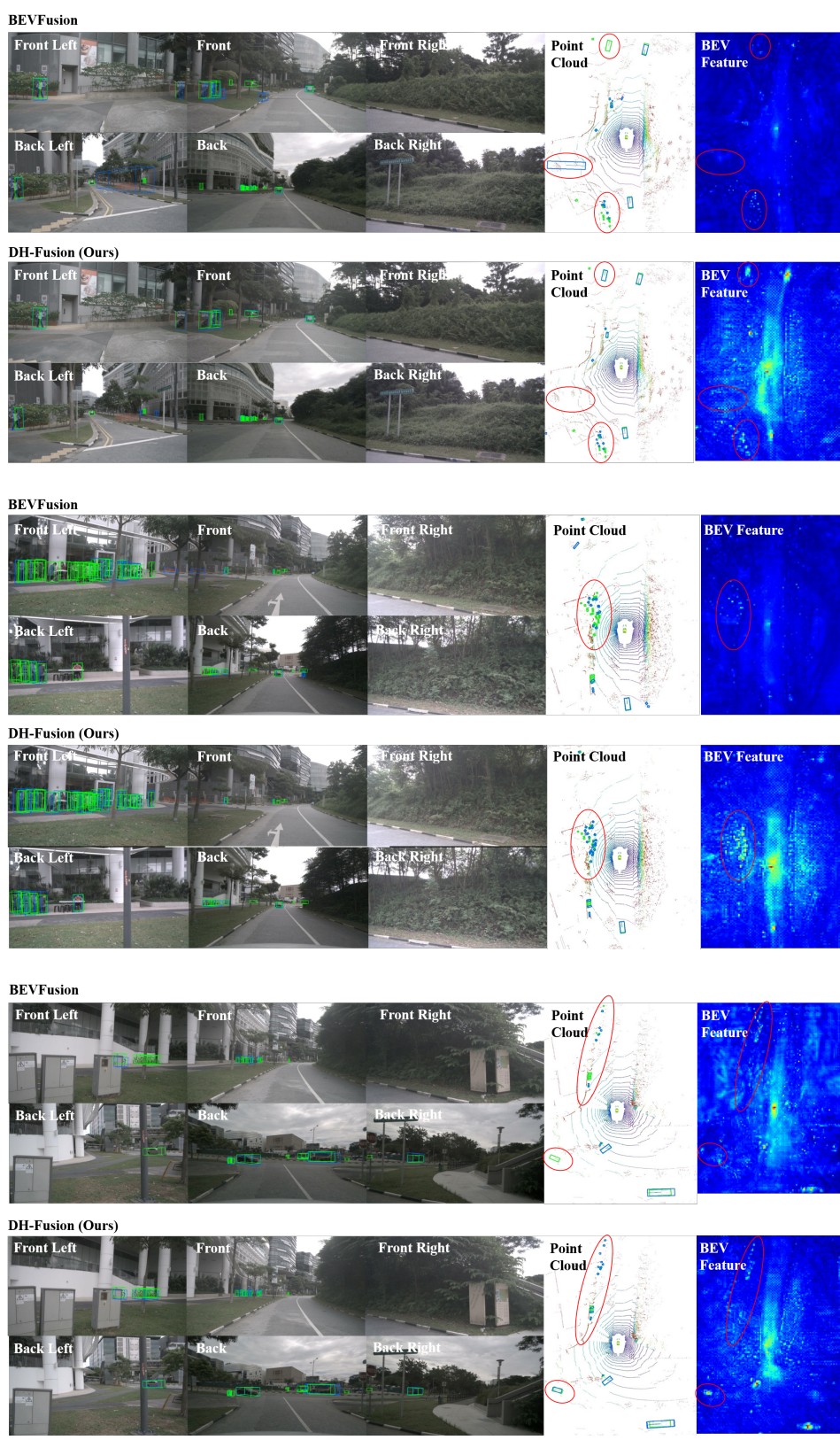

Figure 7: More examples of 3D object detection results and BEV features from BEVFusion and ours. We show the ground truth boxes in green, and the prediction boxes in blue. We use red circles to highlight the comparisons of ours with BEVFusion.

