# OpenReview forum: "DH-Fusion: Depth-Aware Hybrid Feature Fusion for Multimodal 3D Object Detection"
_NeurIPS.cc/2024/Conference — Submitted to NeurIPS 2024_

### Official Review · Reviewer_kgkY · 2024-07-06

**Soundness:** 2
**Presentation:** 3
**Contribution:** 2
**Rating:** 4
**Confidence:** 3

**Summary:**

This study reveals that modalities have varying impacts depending on depth, leading to the proposal of DH-Fusion. This method
dynamically adjusts feature weights using depth encoding, improving multi-modal 3D object detection. Results on nuScenes show DHFusion outperforms prior methods.

**Strengths:**

1. This paper is well-presented. The structure is clear and easy to follow.
2. Comprehensive experiments on the nuScenes dataset are conducted to validate the effectiveness of the proposed DH-Fusion.

**Weaknesses:**

1. Lake of Novelty: The Depth Encoder in DH-Fusion is similar to the 3D Position Encoders in PETR (PETR: Position embedding transformation for multi-view 3d object detection). The Depth-Aware Global Feature Fusion (DGF) module and Depth-Aware Local Feature Fusion (DLF) module in DH-Fusion are analog to the Hierarchical Scene Fusion (HSF) module and Instance-Guided Fusion (IGF) module in IS-Fusion (IS-Fusion: Instance-scene collaborative fusion for multimodal 3d object detection). In conclusion, the contribution of this work seems like "A+B," which is limited.
2. For the nuScenes test leaderboard, DH-Fusion achieved a Top 10 ranking only with 384x1056 image size and SwinTiny backbone.
Please provide results when using larger 900x1600 image size and ConvNeXtS backbone.

**Questions:**

1.Please analyze the theoretical reasons for DH-Fusion's robustness advantage against various corruptions, as in nuScenes-C.
2. In Table 1, for experiments on the nuScenes dataset, It's necessary to include metrics like mATE, mASE, mAOE, mAVE, and mAAE as
done in other papers
3.Do the authors plan to release the code and provide pre-trained, especially nuscenes test leaderboard models for further research?

---

> ### Author Rebuttal · Authors · 2024-08-06
>
> We would like to express our sincere gratitude to you for the valuable comments and constructive feedback. Below, we address each each of question or comment in detail.
>
> **Comment 1:** "Lake of Novelty: The Depth Encoder in DH-Fusion is similar to the 3D Position Encoders in PETR (PETR: Position embedding transformation for multi-view 3d object detection). The Depth-Aware Global Feature Fusion (DGF) module and Depth-Aware Local Feature Fusion (DLF) module in DH-Fusion are analog to the Hierarchical Scene Fusion (HSF) module and Instance-Guided Fusion (IGF) module in IS-Fusion (IS-Fusion: Instance-scene collaborative fusion for multimodal 3d object detection). In conclusion, the contribution of this work seems like "A+B," which is limited."
>
> **Response 1:** We disagree. In this work, we for the first time point out that depth is an important factor to consider during feature fusion by providing insightful analysis. This new finding makes our method differ from previous fusion methods fundamentally. On one hand, our proposed depth encoder is used to adaptively adjust the weights of features from different modalities at different depths; it is conceptually different from 3D Position Encoders in PETR, which are used to relate 2D image features with reference point positions for obtaining 3D features. On the other hand, our proposed DGF and DLF modules perform feature fusion in an adaptive way, i.e., the RGB/point cloud features are assigned with different weights as depth varies; such a dynamic mechanism is novel and has not been used by IS-Fusion or previous fusion methods. Therefore, we believe our method is novel, and the provided insights are inspiring for the community.
>
> **Comment 2:** "For the nuScenes test leaderboard, DH-Fusion achieved a Top 10 ranking only with 384x1056 image size and SwinTiny backbone. Please provide results when using larger 900x1600 image size and ConvNeXtS backbone."
>
> **Response 2:** Thank you for your feedback. We provide a more comprehensive evaluation of the performance of DH-Fusion with different image encoders, including using larger 900x1600 image sizes and the ConvNeXtS backbone. We observe a further improvement while using a larger backbone and image size, indicating the scalability of our method.
>
> **Table 1: Performance with different image encoders**
>
> | Image Encoder | Resolution       | NDS  | mAP  |
> |---------------|------------------|------|------|
> | ResNet18      | 256 × 704        | 73.3 | 69.8 |
> | ResNet50      | 320 × 800        | 74.0 | 71.2 |
> | SwinTiny      | 384 × 1056       | 74.4 | 72.3 |
> | ConvNeXtS     | 900 × 1600       | 74.9 | 72.9 |
>
> **Comment 3:** "Please analyze the theoretical reasons for DH-Fusion's robustness advantage against various corruptions, as in nuScenes-C."
>
> **Response 3:** DH-Fusion’s robustness against various data corruptions in the nuScenes-C dataset is attributed to the combined effects of depth encoding and cross-attention. On one hand, depth encoding is helpful in the way of allowing the model to dynamically weigh features as depth varies. For example, on foggy days, those objects at a far distance are more invisible on RGB images than those at a near distance, and thus the RGB features should be down-weighted at a far distance. On the other hand, the cross-attention mechanism allows DH-Fusion to dynamically focus on the most relevant features and suppress those ineffective features caused by corruptions across modalities at global and local levels. In this way, it reduces the negative impact of corruptions to some extent.
>
> **Comment 4:** "In Table 1, for experiments on the nuScenes dataset, It's necessary to include metrics like mATE, mASE, mAOE, mAVE, and mAAE as done in other papers."
>
> **Response 4:** Thank you for your valuable suggestion. We will provide the results of our method on the metrics mATE, mASE, mAOE, mAVE, and mAAE.
>
> **Table 2: Results on nuScenes test set**
>
> | Methods               | NDS ↑ | mAP ↑ | mATE ↓ | mASE ↓ | mAOE ↓ | mAVE ↓ | mAAE ↓ |
> |-----------------------|-------|-------|--------|--------|--------|--------|--------|
> | DH-Fusion-light (Ours)| 74.2  | 70.9  | 26.1   | 24.3   | 32.4   | 17.8   | **12.2** |
> | DH-Fusion-base (Ours) | 74.7  | 71.7  | 25.2   | 23.6   | 32.9   | 18.5   | 12.7   |
> | DH-Fusion-large (Ours)| **75.4**  | **72.8**  | **24.7**   | **23.2**   | **32.1**   | **17.7**   | 12.5   |
>
> **Table 3: Results on nuScenes validation set**
>
> | Methods               | NDS ↑ | mAP ↑ | mATE ↓ | mASE ↓ | mAOE ↓ | mAVE ↓ | mAAE ↓ |
> |-----------------------|-------|-------|--------|--------|--------|--------|--------|
> | DH-Fusion-light (Ours)| 73.3  | 69.8  | 27.2   | 25.0   | **26.4**   | 17.9   | 18.3   |
> | DH-Fusion-base (Ours) | 74.0  | 71.2  | 26.8   | 24.8   | 27.9   | 17.9   | 18.2   |
> | DH-Fusion-large (Ours)| **74.4**  | **72.3**  | **26.3**   | **24.7**   | 26.5   | **17.8**   | **18.2**   |
>
> **Comment 5:** "Do the authors plan to release the code and provide pre-trained, especially nuScenes test leaderboard models for further research?"
>
> **Response 5:** We appreciate the reviewer's interest in our work and the potential for further research using our models. We plan to release our code and pre-trained models, including those used for the nuScenes test leaderboard, upon the acceptance of our paper.

---

> > ### Comment · Reviewer_kgkY · 2024-08-08
> >
> > Thanks to the authors for the detailed response. The completeness and rigorousness are enlarged with the new experiment results. My dominant concern is still the significance of the contribution, which is also expressed by other reviewers. The universality for 3D detection task and other datasets has not been properly verified, especially when the current performance promotion is not significant against the SOTA methods. I would like to keep my current recommendation.

---

> ### Author Response · Authors · 2024-08-13
>
> Thank you for your continued feedback. We would like to highlight that our method balances both accuracy and efficiency. Specifically, our DH-Fusion-light is the first to achieve over 10 FPS on a RTX 3090 GPU, while maintaining comparable accuracy, thereby meeting the requirements for real-time applications. Additionally, while the latest methods we compared are evaluated solely on the nuScenes dataset, we have further evaluated our methods on nuScenes-C, providing a more comprehensive evaluation. Furthermore, as reviewer qiTv also noted, the datasets we employed are sufficient to demonstrate the effectiveness of our approach.

---

### Official Review · Reviewer_kR8p · 2024-07-09

**Soundness:** 2
**Presentation:** 3
**Contribution:** 2
**Rating:** 5
**Confidence:** 3

**Summary:**

This paper proposed a LiDAR-camera modality feature fusion method based on depth encoding for robust 3D object detection. Based on the observation that the LiDAR and camera modality information should have dynamic relative importance depending on the distance of object to be detected, the paper proposed a Depth-Aware Hybrid Feature Fusion (DH-Fusion) strategy which consists of a Depth-Aware Global Feature Fusion (DGF) module and a Depth-Aware Local Feature Fusion (DLF) module. Experiment on the public nuScenes and nuScenes-C dataset demonstrates that the proposed method is robust to various kinds of corruptions and achieves SOTA performance on 3D object detction.

**Strengths:**

1. The idea of depth-aware multimodality feature fusion for 3D object detection is reasonable, especially for the detection of distant objects.
2. The ablation study clearly demonstrates the effectiveness of the proposed DGF&DLF module when using BEVFusion as baseline
3. The presentation is clear and the ability of the proposed method on the detection of distant object in Figure 6 is impressive

**Weaknesses:**

1. How about the algorithm's performance on small object detection? small object could be normal-sized object at far distance or small-sized object in near distance, is it possible that the proposed depth-aware module hurts the detection performance of small-sized object in near distance? since according to Figure 5, LiDAR modality will have relatively larger weights at near distance, but it is in low resolution, so not good for small object detection.
2. Compare with SOTA, the achieved performance improvement is not that significant. as shown in table 1, the performance gap between the proposed method and IS-Fusion is small and IS-Fusion even achieves slightly better mAP, it is not clear whether the proposed method can achieve similar performance improvement as indicated in ablation study when using IS-Fusion as baseline.
3. In Figure 5(b), it would be good to add a color bar to indicate the magnitude corresponding to each color

**Questions:**

The main concern is on the experiment verification of the proposed method, as listed in the weakness part. I may adjust my rating if such concerns are well addressed

**Limitations:**

There is no paragraph explaining the weakness of the proposed method.

---

> ### Author Rebuttal · Authors · 2024-08-06
>
> We would like to express our sincere gratitude to you for the valuable comments and constructive feedback. Below, we address each of question or comment in detail.
>
> **Comment 1:** "How about the algorithm's performance on small object detection? small object could be normal-sized object at far distance or small-sized object at near distance, is it possible that the proposed depth-aware module hurts the detection performance of small-sized object at near distance? since according to Figure 5, LiDAR modality will have relatively larger weights at near distance, but it is in low resolution, so not good for small object detection."
>
> **Response 1:** Thank you for raising this important point. To address your concerns, we consider cars as normal-sized objects, and pedestrians, motorcycles, and bicycles as small-sized objects. We conduct experiments to evaluate our method on normal-sized objects at far distance and small-sized objects at near distance. For these above small objects, our method outperforms the state-of-the-art method IS-Fusion, as well as our baseline BEVFusion, demonstrating our robustness for small object detection. It is true that LiDAR modality has larger weights at near distance, as it is in high resolution as shown in Fig. 1(b) of the original submission, so it does not hurt but in fact helps small-sized object detection at near distance.
>
> **Table 1: Performance on small objects, including normal-sized objects at far distance (>30m) and small-sized objects at near distance (0-20m). The numbers are AP.**
>
> | Methods               | >30m | | | 0-20m| |
> |-----------------------|------|-|------------|------------|---------|
> |                           | Car | | Pedestrian | Motorcycle | Bicycle |
> | BEVFusion             | 72.1 | | 92.9       | 89.9       | 75.7    |
> | IS-Fusion             | 76.1 | | 94.1       | 90.2       | 78.4    |
> | DH-Fusion-large (Ours)| **77.2** | | **94.2** | **91.5** | **78.6** |
>
> **Comment 2:** "Compare with SOTA, the achieved performance improvement is not that significant. As shown in Table 1, the performance gap between the proposed method and IS-Fusion is small and IS-Fusion even achieves slightly better mAP, it is not clear whether the proposed method can achieve similar performance improvement as indicated in ablation study when using IS-Fusion as baseline."
>
> **Response 2:** Thank you for your feedback. To address the concern, we conduct an experiment using IS-Fusion as the baseline, and integrate our depth encoder into its IGF module, which allows to adjust the weights of image features with depth during instance feature fusion. We note that our method still achieves improvements when applied to a stronger baseline. This demonstrates the generalization ability of our approach across different baselines. We plan to dedicate more time to refining our experiments in future work to achieve more significant performance improvements.
>
> **Table 2: Ablation studies using IS-Fusion as the baseline.**
>
> | Methods   | NDS                 | mAP               |
> |--------------|----------------------|--------------------|
> | IS-Fusion | 73.6                  | 72.5                |
> | w/ DE     | **74.1 (+0.5)**   | **72.7 (+0.2)** |
>
> **Comment 3:** "In Figure 5(b), it would be good to add a color bar to indicate the magnitude corresponding to each color."
>
> **Response 3:** Thank you for your valuable suggestion. We will add a color bar to Figure 5(b) in the final version of the paper to indicate the magnitude corresponding to each color.
>
> **Comment 4:** "There is no paragraph explaining the weakness of the proposed method."
>
> **Response 4:** We apologize for the oversight. We condense the discussion of our method's limitations into the conclusion due to limited space, and do not dedicate a separate section to it. We will address this in the final version of the paper.

---

> > ### Comment · Reviewer_kR8p · 2024-08-13
> >
> > Thanks the authors for their detailed response with additional experiment, Most of my concerns have been addressed. So I will increase my rating

---

> ### Comment · Area_Chair_Teu3 · 2024-08-13
>
> Dear Reviewer kR8p
>
> Thanks for reviewing this work. Would you mind to check authors' feedback and see if it resolves your concerns or you may have further comments?
>
> Best wishes
>
> AC

---

### Official Review · Reviewer_qiTv · 2024-07-10

**Soundness:** 3
**Presentation:** 3
**Contribution:** 3
**Rating:** 7
**Confidence:** 4

**Summary:**

This paper introduces a novel strategy for LiDAR-camera 3D object detection that emphasizes the importance of depth information in feature fusion processes. The authors argue that different modalities, such as LiDAR point clouds and RGB images, contribute variably at different depths, and this variation has been overlooked in previous works. The key contribution is the Depth-Aware Hybrid Feature Fusion (DH-Fusion) strategy that dynamically adjusts the weights of point cloud and image features based on depth encoding at both global and local levels. The DH-Fusion method surpasses previous state-of-the-art methods in terms of NDS on the nuScenes dataset and demonstrates robustness to various data corruptions. In general, the design is reasonable and performance is impressive.

**Strengths:**

1.	The paper is well-structured, with a clear abstract, introduction, methodology, experiments, and conclusion sections that logically flow from one to the next.
2.	The authors effectively communicate complex ideas through clear language and comprehensive illustrations, aiding the reader's understanding of the proposed method.
3.	The motivation of design is clear and experiments are extensive.
4.	The idea of depth encoding for dynamical fusion is interesting and reasonable.
5.	The performance is very impressive and the robustness makes the method more applicable to challenging scene.

**Weaknesses:**

The paper has no obvious weakness except they didn't do experiments on other datasets.
But I think the nuScenes is already large enough to demonstrate the general effectiveness.

**Questions:**

I noticed this paper encodes depth information using cosine functions, but I haven't seen experiments validating the impact of cosine functions. Would there be a significant performance drop if distances were used directly instead of cosine functions?

**Limitations:**

There is no discussion of limitation in main text, but a justification is given in Checklist: using an attention-based approach to interact with the two modalities makes the detection results sensitive to modality loss.

---

> ### Author Rebuttal · Authors · 2024-08-06
>
> We would like to express our sincere gratitude to you for the valuable comments and constructive feedback. Below, we address each question in detail.
>
> **Comment 1:** "I noticed this paper encodes depth information using cosine functions, but I haven't seen experiments validating the impact of cosine functions. Would there be a significant performance drop if distances were used directly instead of cosine functions?"
>
> **Response 1:** We appreciate the reviewer's question regarding the use of cosine functions for encoding depth information. We conduct the experiments using normalized depth directly as the depth encoding in our feature fusion module, without applying cosine functions. Our experimental results in the table below show a performance drop when using normalized depth directly. We believe that depth encoding benefits from the use of cosine functions to capture the periodicity and symmetry of the depth information relative to the ego vehicle. The cosine function helps in better representing the variations in depth, leading to model performance improvement.
>
> **Table 1: Ablation studies of cosine functions.**
>
> | Methods                | NDS             | mAP             |
> |------------------------|-----------------|-----------------|
> | Baseline + DGF          | 72.4            | 69.4           |
> | w/o cosine functions   | 72.1 (-0.3)   | 68.5 (-0.9) |
> | Baseline + DLF           | 72.7            | 69.3           |
> | w/o cosine functions   | 72.3 (-0.4)   | 68.6 (-0.7) |
>
> **Comment 2:** "There is no discussion of limitation in main text, but a justification is given in Checklist: using an attention-based approach to interact with the two modalities makes the detection results sensitive to modality loss."
>
> **Response 2:** We apologize for the oversight. We condense the discussion of our method's limitations into the conclusion due to limited space, and do not dedicate a separate section to it. We will address this in the final version of the paper.

---

> > ### Comment · Reviewer_qiTv · 2024-08-08
> >
> > After thoroughly reviewing the feedback from reviewers and the author's responses, I've noted that some reviewers advocate for further validation on additional datasets. However, I think that the conducted experiments on the nuScenes and nuScenes -C datasets provide enough evidence to substantiate the efficacy of the proposed method. The author has addressed my concerns, and as such, I maintain my original recommendation without alteration.

---

### Official Review · Reviewer_tCBR · 2024-07-13

**Soundness:** 2
**Presentation:** 3
**Contribution:** 2
**Rating:** 4
**Confidence:** 5

**Summary:**

The paper introduces DH-Fusion, a novel Depth-Aware Hybrid Feature Fusion strategy for multimodal 3D object detection that leverages LiDAR and camera data. The key innovation lies in dynamically adjusting the weights of point cloud and RGB image features based on depth encoding at both global and local levels. The authors propose two modules: Depth-Aware Global Feature Fusion (DGF) and Depth-Aware Local Feature Fusion (DLF), which enhance feature integration and compensate for information loss during the transformation to Bird's-Eye-View (BEV) space. Experiments on the nuScenes dataset demonstrate that DH-Fusion surpasses state-of-the-art methods in terms of Novelty Detection Score (NDS) and is more robust to data corruptions, as evidenced by superior performance on the nuScenes-C dataset.

**Strengths:**

1. The paper proposes a novel feature fusion strategy that adaptively adjusts the weights of LiDAR point cloud and RGB image features based on depth
2. The introduction of depth encoding at both global and local levels allows for more nuanced and context-aware feature integration, enhancing the detector's ability to understand the scene's depth structure.

**Weaknesses:**

1. The authors only present results on nuScenes dataset. The alogrithms should be also evaluated on other prevailing public dataset like KITTI.
2. The depth-aware fusion might be tailored to the specific characteristics of the training dataset, potentially leading to overfitting and reduced performance on diverse or unseen data.
3. While the paper includes ablation studies, a more extensive set of experiments that isolate the impact of different components of the system could provide deeper insights.

**Questions:**

What is the computational complexity of the DH-Fusion model, and how does it compare with other state-of-the-art methods in terms of runtime and resource usage?

**Limitations:**

1.  While the method shows strong performance on the nuScenes dataset, its generalizability to other datasets or varied real-world conditions might require further investigation.
2. The paper does not provide a detailed discussion on the computational efficiency, which is crucial for practical applications, especially in terms of processing time and resource usage.
3 .The method assumes high-quality, synchronized data from LiDAR and camera sensors, which might not always be guaranteed in real-world scenarios.

---

> ### Author Rebuttal · Authors · 2024-08-06
>
> We would like to express our sincere gratitude to you for the valuable comments and constructive feedback. Below, we address each of question or comment in detail.
>
> **Comment 1:** "What is the computational complexity of the DH-Fusion model, and how does it compare with other state-of-the-art methods in terms of runtime and resource usage?"
>
> **Response 1:** Thank you for your question. The parameters of our DH-Fusion-light, DH-Fusion-base and DH-Fusion-large are 40.38M, 53.05M, and 56.94M, respectively. The flops of these models are 271.6G, 822.8G, 1508.2G, respectively. The runtime of these models are 72.46ms, 114.94ms, and 175.44ms on a 3090 GPU, respectively. In Table 1 of the original submission, we compare our method with other SOTA methods in terms of FPS. Specifically, under the same configuration, our DH-Fusion-light runs faster than BEVFusion, and achieves a real-time inference speed; our DH-Fusion-base maintains comparable inference speed, compared to FocalFormer3D; our DH-Fusion-large runs 2x faster than IS-Fusion.
>
> **Comment 2:** "The authors only present results on nuScenes dataset. The algorithms should be also evaluated on other prevailing public dataset like KITTI; While the method shows strong performance on the nuScenes dataset, its generalizability to other datasets or varied real-world conditions might require further investigation."
>
> **Response 2:** In our original submission, we have actually presented results on two datasets: the nuScenes dataset and the nuScenes-C dataset. The detailed results can be found in Table 1 and Table 2. In particular, our experiments on the nuScenes-C dataset, which includes various realistic noise conditions, show that our method exhibits high robustness under diverse real-world corruption conditions and very good generalization ability. Since the KITTI dataset provides only stereo images instead of multi-view images, our method cannot be directly applied on it, and we believe that the nuScenes dataset, providing 700 different scene sequences for training, 300 scene sequences for validation and testing, is sufficiently large and diverse, serving as a standard benchmark in the field of 3D object detection. The methods we compared [1, 2] against typically conduct their experiments solely on this dataset as well.
>
> [1] Liu, Z., Tang, H., Amini, A., Yang, X., Mao, H., Rus, D.L., Han, S.: Bevfusion: Multi-task multi-sensor fusion with unified bird’s-eye view representation. In: ICRA (2023)
>
> [2] Yin, J., Shen, J., Chen, R., Li, W., Yang, R., Frossard, P., Wang, W.: Is-fusion: Instance-scene collaborative fusion for multimodal 3d object detection. In: CVPR (2024)
>
> **Comment 3:** "The depth-aware fusion might be tailored to the specific characteristics of the training dataset, potentially leading to overfitting and reduced performance on diverse or unseen data."
>
> **Response 3:** No, our method is not tailored to any dataset. Specifically, our depth-aware fusion method adaptively adjusts the weights of different modalities based on depth, which is inherently independent of the specific dataset used for training. We acknowledge that the performance of our method on the nuScenes-C dataset, where we achieve 68.67 NDS and 63.07 mAP, shows a decrease compared to the nuScenes dataset. The reduced performance on nuScenes-C can be attributed to the increased difficulty of this dataset, which includes various corruptions. Despite this, our method still outperforms other approaches, demonstrating its robustness.
>
> **Comment 4:** "While the paper includes ablation studies, a more extensive set of experiments that isolate the impact of different components of the system could provide deeper insights."
>
> **Response 4:** We appreciate the reviewer's suggestion for a more extensive set of experiments to isolate the impact of different components of the system. In fact, we have already provided a detailed discussion on the impact of various components on the model in the paper. Specifically, section 4.4 thoroughly examines the influence of different components, including the effect of DGF and DLF, and the effect of the depth encoding.

---

> ### Comment · Area_Chair_Teu3 · 2024-08-13
>
> Dear Reviewer tCBR
>
> Thanks for reviewing this work. Would you mind to check authors' feedback and see if it resolves your concerns or you may have further comments?
>
> Best wishes
>
> AC

---

### Author Rebuttal · Authors · 2024-08-06

We thank all reviewers for their valuable comments and constructive suggestions, and are glad they appreciate that "The paper proposes a novel feature fusion strategy that adaptively adjusts the weights of LiDAR point cloud and RGB image features based on depth" (Reviewer tCBR), "The idea of depth encoding for dynamical fusion is interesting and reasonable" (Reviewer qiTv), "The idea of depth-aware multimodality feature fusion for 3D object detection is reasonable, especially for the detection of distant objects" (Reviewer kR8p), and "Comprehensive experiments on the nuScenes dataset are conducted to validate the effectiveness of the proposed DH-Fusion." (Reviewer kgkY).

Our paper presents a novel approach, Depth-Aware Hybrid Feature Fusion (DH-Fusion), for multi-modal 3D object detection. This method leverages depth encoding to adaptively adjust feature weights during fusion, which significantly enhances detection performance. We highlight the following key contributions of our work:

-  To the best of our knowledge, we for the first time identify depth as a crucial factor in the fusion of LiDAR point cloud and RGB image features for 3D object detection. Our statistical and visualization analyses reveal that the role of image features varies with depth, emphasizing the need for depth-aware adjustments in feature fusion.

-  We propose a depth-aware hybrid feature fusion strategy that dynamically adjusts feature weights at both global and local levels by integrating depth encoding. This strategy comprises the Depth-Aware Global Feature Fusion (DGF) module and the Depth-Aware Local Feature Fusion (DLF) module. The DGF module utilizes a global-fusion transformer encoder with depth encoding to adaptively the weight of image BEV features, while the DLF module refines local instance features by utilizing the original instance features using a local-fusion transformer encoder with depth encoding. This approach ensures high-quality feature extraction and optimal utilization of multi-modal data across varying depths.

- Our method has been evaluated on the nuScenes dataset and the more challenging nuScenes-C dataset. The results demonstrate that DH-Fusion not only outperforms previous multi-modal methods but also maintains robustness against various types of data corruption. This highlights the effectiveness and reliability of our proposed method in real-world scenarios.

Considering the valuable feedback provided by all reviewers, we conduct additional experiments and provide detailed results in PDF:

- Table 1: Ablation studies of cosine functions.
- Table 2: Performance on small objects, including normal-sized objects at far distance (>30m) and small-sized objects at near distance (0-20m).
- Table 3: Ablation studies using IS-Fusion as the baseline.
- Table 4: Performance with different image encoders.
- Table 5: More detailed results on nuScenes test set.
- Table 6: More detailed results on nuScenes validation set.

We hope additional experiments address the concerns raised and further validate the effectiveness of our proposed method, which will be included in the supplementary materials.

Finally, we believe our method offers useful insights for feature fusion in the field of multi-modal 3D object detection. We hope our contributions will inspire further research and development in this area.

---

### Decision · Program_Chairs · 2024-09-25

**Decision:**

Reject

**Comment:**

This paper proposed a LiDAR-camera modality feature fusion method by introducing the depth encoding at both global and local levels for 3D object detection.
Concerns are about results on different dataset, ablation studies, and the limited performance improvement.
The authors argue that the latest published methods are also evaluated solely on the nuScenes dataset.
The reviewers think this point is not significant enough for the proposed method to be accepted.
The AC recommends rejection for this submission.